# Effect of Pre-Anodized Film on Micro-Arc Oxidation Process of 6063 Aluminum Alloy

**DOI:** 10.3390/ma15155221

**Published:** 2022-07-28

**Authors:** Linwei Li, Erhui Yang, Zhibin Yan, Xiaomeng Xie, Wu Wei, Weizhou Li

**Affiliations:** 1School of Resources, Environment and Materials, Guangxi University, Nanning 530004, China; llinwei888@163.com (L.L.); 89yangeh@sina.com (E.Y.); 202010103809@mail.scut.edu.cn (Z.Y.); x648033@163.com (X.X.); 57226028@163.com (W.W.); 2School of Materials Science and Engineering, Xiamen University of Technology, Xiamen 361024, China; 3School of Mechanical and Marine Engineering, Beibu Gulf University, Qinzhou 535011, China; 4School of Materials Science and Engineering, South China University of Technology, Guangzhou 510640, China

**Keywords:** MAO, pre-anodized, microstructure, energy consumption

## Abstract

In the current investigation, micro-arc oxidation (MAO) ceramic coatings on aluminum are galvanostatically synthesized at various processing stages in an alkaline silicate system. The resultant coatings are systematically investigated in terms of the following respects: The working voltage and surface sparking evolution over the studied course of MAO are recorded by the signal acquisition system and the real-time imaging, respectively; the phase composition, the surface morphology, and the polished cross-section of the coatings are characterized by X-ray diffraction (XRD) and scanning electron microscopy (SEM) assisted with an energy-dispersive X-ray spectrometer (EDS), respectively. In particular, with the help of a low-rate increase in working voltage, the evolution of the sparks, the energy consumption, and the microstructure development of aluminum in alkaline silicate electrolyte by pre-anodizing are systematically investigated. The results show that the pre-anodized film can accelerate the evolution process of MAO spark and shorten the reaction process in the early stage of MAO reaction, reducing energy consumption and improving the corrosion resistance of the MAO coating. The γ-Al2O3 phase content after pre-anodized is significantly increased in MAO coatings. In particular, the thicker the pre-anodized film (beyond 8 μm) was broken down and fragmentation thinning in the early stage of the MAO process with the presence of micro discharges. This is due to the fact that the electron transition will be released by the emission of radiative recombination and reveals obvious galvanoluminescence (GL) behavior on the surface of the pre-anodized film. Further, based on the present MAO coating microstructure, a model of coating growth after pre-anodized that evolves over time is proposed.

## 1. Introduction

Micro-arc oxidation (MAO), also called plasma electrolytic oxidation (PEO), is a promising surface treatment technique that could produce ceramic-like coatings, to improve corrosion and wear resistance on aluminum alloys by using environment-friendly dilute electrolytes [1,2]. The growth of the MAO coating is a complicated reaction process, accompanied by instantaneously and micro-locally sonic, light and heat emission, chemical and plasma-chemical reactions, and a melt and quencher [3]. Nowadays, much research is on changing the composition of the electrolyte and electrical parameters to increase the MAO coating of corrosion and wear resistance [4,5] or obtain functional coatings with bio-related performance [6,7], catalysis [8,9], and light-related performance [10,11]. So, the MAO technique change has been gradually applied in aerospace, automobile, medical treatment, marine protection, and other fields, and its application field is expanding [3,12,13]. However, MAO technology has high energy consumption and low energy utilization, for example, 10–60 A/dm^2^ and low coating efficiency (~25%) [14,15], which hinders its further development and application [16,17].

At present, the ways of MAO energy saving include (1) regulation of power supply parameters: Gebarowski and Pietrzyk et al. [18] showed that by adjusting the Rpn (the ratio of positive charge to negative charge) value, the transition to a soft plasma state occurs when the R_pn_ value is between 0.8 and 0.9, which makes the reaction process transition to soft plasma with reductions in energy savings of up to ~23%. Or adjust the power parameters (duty cycle, cathode spacing, unipolar or bipolar MAO mode) to increase the coating growth rate and reductions in energy savings [19,20]. (2) Electrolyte design: Adopt different electrolyte components to increase the conductivity of the solution and reduce the breakdown voltage [21], or add particles to the electrolyte to reductions in energy savings in the reaction process [22]. (3) Pre-anodized technology: the pre-anodized film on aluminum alloys has been demonstrated to promote the establishment of a micro-arc regime, accelerate the growth of the MAO coating and participate in the growth of the MAO coating at the same time [23], which significantly reduces energy consumption. Pre-anodized aluminum alloys save up to 50% energy compared to conventional micro-arc oxidation treatments [24,25]. Mohedanoa et al. [26] achieved an energy consumption of 3.2 kW·h·m-2·μm-1 for MAO coating based on electrolyte selection and precursor anode film. Energy consumption is significantly reduced compared to conventional treatment. However, the studies about pre-anodized technology are quite recent. There is a lack of research on the transformation process from pre-anodized film to MAO coating. So there is a need to investigate the changes in the spark, film morphology, and growth mechanism. In fact, to the best of the authors’ knowledge, there is no data available regarding the spark change and growth mechanism of the pre-anodized process of micro-arc oxidation (MAO).

In this work, with the help of a low-rate increase in working voltage, the evolution of the sparks, the energy consumption, the microstructure development, and the corrosion resistance of aluminum in alkaline silicate electrolyte by pre-anodizing are systematically investigated.

## 2. Experimental Methods

### 2.1. Materials

6063 aluminum alloy specimens with a dimension of 20 mm × 15 mm × 3 mm were used as the substrate material in the experiments. All specimens were polished with SiC sandpaper from 400 to 1500 grit, and then ultrasonically cleaned in ethanol.

### 2.2. Surface Treatment Based on Anodizing Techniques

The pre-anodization was prepared by a DC power supply. The samples pre-anodized with a thickness of 8 μm, 16 μm and 24 μm respectively were formed in sulfuric acid solution (150 g/L) during different times of 20 min, 40 min and 60 min respectively at 10 A/dm^2^ and ~25 °C.

The MAO coatings were prepared by a pulse power source (homemade pulsed power) with a 10 A/dm^2^ square waveform at 600 Hz, 30% duty cycle. A dilute alkaline electrolyte with the addition of 10 g/L Na_2_SiO_3_ and 2 g/L NaOH is used. The sample was used for the working electrode (anode), and the stainless-steel electrode plate acted as the cathode. During the treatment, the electrolyte temperature is kept below 50 °C by using a stirring and cooling system. After MAO treatment, the samples pre-anodized with a thickness of 8 μm, 16 μm and 24 μm were named A1, A2, A3 respectively. The without pre-anodized samples were named A0. Additionally, all the samples were cleaned with deionized water and dried at ambient temperature.

### 2.3. Coating Characterizations

The real-time images of the micro-discharges occurring on MAO specimens were monitored by a commercial digital camera (Handycam FDR-AX100, SONY, China) with a frame rate of 100 fps over the MAO progress. The thickness of the coating was measured by the TT260B coating eddy current gauge. The average value was calculated after 10 measurements at different positions of the film surface. The morphologies, cross-sections, and element content of specimens were examined after standard metallographic preparation by scanning electron microscopy using an FE-SEM/EDS SU-8020/X-MAX 80 instrument (Hitachi-High-Technologies-Corporation, Tokyo, Japan) equipped with energy-dispersive Xray (EDS) microanalysis hardware. The D/MAX 2500V X-ray diffraction (XRD, Rigaku, Tokyo, Japan) was used to detect the phase structure of the coatings with a scan speed of 6°/min, a 2θ range of 20° to 80°, a working voltage of 40 kV and a current of 150 mA.

### 2.4. Specific Energy Consumption

The specific energy consumption is calculated by integration of the voltage-time and current-time response signal by the signal acquisition system of power supply during the treatment and according to Equation (1). The total energy (ρ) is the energy consumed during the MAO process and the anodizing process. Results were calculated as an average of four samples.
(1)$$\rho =\frac{\int_{t0}^{T_{0}}V_{0}\cdot I_{0}dt}{Sd360}$$
where *V*_0_ is instantaneous voltage, *I*_0_ is instantaneous current density, *t*_0_ is the starting point of the experiment, and *T*_0_ is the end time of the experiment of pre-anodized. *S* is the sample surface area (cm^2^), and *d* is the coating thickness (μm).

### 2.5. Corrosion Performance

The samples were placed separately in beakers containing 30 mL 3.5 wt% NaCl solution for 168 h to analyze the corrosion resistance. The soaked samples were washed in deionized water, dried and then weighed. The samples with the lowest corrosion weight loss and specific energy consumption were selected for the potentiodynamic polarization scanning measurements. The potentiodynamic polarization scanning measurements of the samples were carried out by an electrochemical workstation (CS350, CorrTest) in a 3.5 wt% NaCl solution to further evaluate the corrosion resistance. During the test, 1 cm^2^ of specimens were used as the working electrode, and the saturated calomel electrode and platinum foil were used as the reference and the counter electrodes, respectively. The scan rate was set as 10 mV/s. The measurement results were fitted by the CS Studio5 software (v5.4.627.16, Wuhan Corrtest Instruments Co., Ltd, Wuhan, China).

## 3. Results

### 3.1. Voltage Behavior and Discharges Evolution

Figure 1 shows the voltage and current response curves and the variation of spark discharge characteristics with oxidation time for A0, A1, A2 and A3. The three typical stages can be distinguished during the MAO coating formation process [27], combining the obtained voltage and current response curves with the experiment phenomena (visual observation): the fast-rising stage, the transition stage, and the stabilization stage.

From Figure 1a, the voltage without pre-anodized A0 rises linearly and rapidly in stage I. After the working current reaches the presupposed value and remains stable (Figure 1b, I-T curve), since the energy supply for plasma discharge tends to be stable, the number of plasma discharges increases first and then reaches the rough saturation [3]. Therefore, the U-t curve of A0 decreases tardily and gradually becomes smooth and steady in stage II. In stage III, the voltage of A0 reveals a slow boost rate. However, from Figure 1a, the samples of pre-anodized A1, A2 and A3 were different from the samples without pre-anodized A0. In stage II, the samples that were pre-anodized revealed a faster rise voltage than those without pre-anodized. In addition, the pre-anodization reveals a higher working voltage than without the pre-anodized in stage III. The thicker the pre-anodized, the higher the working voltage. This may be due to the fact that the pre-anodized film replaces the oxide film formed in the traditional MAO process and has higher resistance. The dielectric breakdown of the MAO reaction is caused by electron ionization avalanches by electron injected [28,29]. Because the pre-anodized film with a higher impedance can store more electrons, delaying the electrons ionization avalanches and promoting the favorable micro-arc regime for the growth of the MAO coatings. At the same time, the working voltage is raised to a higher level. The samples A2 and A3 have a similar working voltage. This may be related to their thicker films.

From Figure 1b, for A0, it was without discharge sparks that appeared on the surface in stage I. Then, the surface of A0 reveals a white fine spark in stage II. With the increase in the voltage, the white sparks change to light yellow sparks gradually. In stage III, the sparks gradually changed from light yellow to dark yellow, increased in size, and decreased in number in stage III. It can be observed that the spark from stage II to III is not as extremely obvious as that from stage I to II. It is worth noting that the spark change of pre-anodized was different from that without pre-anodized. From Figure 1b, the pre-anodization shows that the spark appears earlier and the time from white to yellow was shorter than that without pre-anodization. In addition, some scattered yellow sparks appear on the surface of the pre-anodized in stages I and II (Figure 1b, 60 s~110 s, A1, A2 and A3) when the working voltage approaches the breakdown voltage, which is called the galvanoluminescence (GL) phenomenon in some research [30,31]. The GL phenomenon effect could be obvious and detectable to the naked eye. Its intensity increases with the working voltage. It is more obvious with the thickness of pre-anodized. Moreover, the GL behavior was obtained more obviously than in other studies [32,33,34]. This GL phenomenon can be attributed to the radiative recombination of electrons at defects in pre-anodized films. The defects include microcracks, cracks, localized domains of different compositions and impurities, etc. [31]. When the working voltage approaches the breakdown voltage, the film of pre-anodized stores too many electrons and the electronic transitions will be released by the emission of radiative recombination [35]. Additionally, the transitions in the aluminum-oxide–electrolyte system are derived from the following atoms: Al, O, H, Na and C, which are present in electrolytes [36].

It takes about 8 min without pre-anodization to reveal a dark yellow spark, while pre-anodization only needs about 4 min, indicating the pre-anodized film can greatly speed up the process of sparks. At the same time, the evolution of the pre-anodized sparks was similar to the without pre-anodized in stage III, indicating that the pre-anodized film mainly affects stages I and II. We selected a specific time point to study, understand and speculate on the growth and microstructure evolution of the MAO coating. They are 1 min (stage I), 2 min (stage II), 5 min (stage II), 10 min (stage III) and 20 min (stage III) respectively.

### 3.2. Coating Thickness and Efficiency Analysis

Figure 2 shows the variation curves of the coating thickness of diverse samples. It could be found that the coating thickness of each sample increased gradually with time. Moreover, the coatings of pre-anodized were thicker than those without pre-anodization. Obviously, the pre-anodization A2 has the most thickness. In addition, the growth rate of A2 coating was faster than other coatings, indicating the thickness (16 μm) of pre-anodized was best for MAO growth. The initial thickness of the pre-anodized A2 and A3 was 16 μm and 24 μm, respectively. However, the film thickness of A2 and A3 was down to 7 μm and 8 μm, respectively, after 5 min of oxidation. This reduction in thickness can be attributed to the fact that the thicker thickness of pre-anodized leads to stronger radiative recombination of electrons, and the initial film of pre-anodized will be locally lost from the aluminum surface in the presence of damaging micro discharges [23]. Table 1 shows the unit energy consumption of the MAO coatings. Compared with the samples without pre-anodized, the samples with pre-anodized can significantly reduce energy consumption. From Table 1, the A2 has the lowest unit energy consumption after oxidation for 10 min (2.78 kW·h·m^−2^ μm^−1^, ~47 %), indicating that A2 (16 μm) was the best pre-anodized in three kinds of thickness of the pre-anodized film for reducing consumed energy.

### 3.3. Phase Composition

Figure 3 illustrates the XRD of the MAO coated specimens in the silicate electrolyte with various MAO treatment times. The diffraction peaks of the aluminum substrate are present in all samples, and the corresponding XRD spectral intensities progressively reduce with the thickening of the MAO coating. Moreover, the main phase of all sample coatings was γ-Al_2_O_3_, but the content of γ-Al_2_O_3_ phases significantly increased in A2 and A3 compared with A0 and A1 at the same time. In addition, A2 and A3 have a similar content of γ- Al_2_O_3_ phases (Figure 3c,d). This may be due to the increase in energy after pre-anodizing, which enhanced the local transient pressure as well as the temperature to reach the conditions for the formation of γ-Al_2_O_3_ phases [37]. In addition, the diffraction intensity of all samples increased with time. This finding was related to the discharge intensity of the samples. Moreover, stronger discharges caused a violent plasma electrochemical reaction, leading to abundant molten oxide spraying, flowing, and solidifying, increasing the coating thickness and phase peak strength finally. It is worthy of note that all MAO coatings in the present study show the absence of diffraction peaks of α-Al_2_O_3_, which often appears in the MAO coatings formed on aluminum alloys in alkaline silicate solutions in most cases [38]. It may be that the content of α-Al_2_O_3_ in the coatings is too small for XRD to test. Because it is too small a number of allotropic changes from metastable γ-Al_2_O_3_ to stable α-Al_2_O_3_ appear in these coatings in the MAO process [30]. In addition, this phase transformation may not have taken place in these coatings during the MAO process.

### 3.4. Surface Morphology

The surface morphology evolution of diverse coatings during the MAO process is revealed in Figure 4, Figure 5, Figure 6 and Figure 7, respectively. The associated EDS analysis data at some detected points is listed in Table 2 and Table 3.

As shown in Figure 4, after 1 min oxidation, the surface without pre-anodized A0 was flat and some wear marks formed in the preparation process were left on the surface. A large number of small holes appeared on the coating surface after 2 min oxidation and the holes gradually grew (Figure 4b–d), which was due to the increase in spark discharge intensity (Figure 1b). The pancake-like with open pancake-like regions and sponge-like structures appeared on the surface after between 5 and 10 min of oxidation. After between 15 and 20 min of oxidation (Figure 4e,f), the pancake-like with an open pancake-like region disappeared, revealing a closed concave flat area on the surface. The existence of concave flat areas shows that the discharge intensity is high at this moment, and the results of the change of pancake-like regions and concave flat areas with processing time are consistent with the results reported in the literature [39].

The surface morphology evolution of pre-anodized was different from that without pre-anodized in stages I and II. The pre-anodization can make the reaction reach a higher degree in a short time, mainly focusing on shortening the reaction process in the early stage of MAO and making it reach the later stage of reaction faster. However, the thicker pre-anodized will be broken down and reveal a broken state by GL and micro-arc spark working together. After 1 min of oxidation, some sparks breakdown appeared on the surface of the pre-anodized (Figure 5, Figure 6 and Figure 7a, red-coils), which was called the GL breakdown, with obvious luminescence (Figure 1b, A2 and A3), caused by the radiation recombination of electrons at defects in the anodic film [36]. It is noteworthy that the spark breakdown marks of the thicker A2 and A3 were more obvious than the thinner A1 associated with the thicker film of the pre-anodized A2 and A3 storing too many electrons. After 2 min of oxidation, the micro-arc sparking already happened this time, and the surface of pre-anodized showed more obvious MAO breakdown marks (Figure 5, Figure 6 and Figure 7b, yellow-coils) than without pre-anodized. Meanwhile, the surface of the thicker A2 and A3 is accompanied by a broken state. This is due to the thicker pre-anodized having a higher voltage, which is accompanied by higher temperature sparks, resulting in fast evaporation of oxide and electrolyte when the breakdown voltage for the pre-anodized film has been reached [2]. Additionally, the surface of A1 only reveals spark breakdown marks, not a broken state (Figure 5b, yellow coils). It may be related to the A1 with a thinner pre-anodized film. After 10 min oxidation, the surface of the pre-anodized has shown closed concave flat areas faster than without pre-anodized (after 15 min oxidation), indicating that the pre-anodization can accelerate the evolution of surface morphology. It is noteworthy that the gray nodular aluminum–silicon melt and the closed concave flat areas occupied the surface of the A1 after 10 min of oxidation (Figure 5d–f). However, for the A2 and A3, the closed concave flat areas mainly occupied the whole surface coating surfaces after 10 min oxidation (Figure 6 and Figure 7).

The pre-anodization has a similar elemental composition to the without pre-anodized, as shown in Table 2. In addition, the elemental composition level of diverse samples in Table 3 proves that the pre-anodized can accelerate the evolution of surface morphology at the same time. As one of the microstructure defects of MAO coating, cracks were found on all samples, from Figure 4, Figure 5, Figure 6 and Figure 7. The pre-anodization has more cracks than those without pre-anodized. The thicker the pre-anodized film, the more cracks. The cracks are closely related to the internal stress relaxation caused by the solidification and shrinkage process of the liquid coating material in the ceramic coating [40]. The pre-anodization has a higher working voltage that is accompanied by more energy, resulting in violent reactions and increases cracks.

### 3.5. Polished Cross-Section Morphology

Figure 8 shows the polished cross-sectional morphology evolution of the MAO coating of different samples in alkaline silicate electrolytes with the MAO treatment time.

As shown in Figure 8, the sample of A0, it can be seen that the coating thickness was increased with oxidation time and that both the coating surface and coating/substrate interface outlines were gradually bent. The cavities defect was revealed after 5 min oxidation, the size of the cavity was increased with oxidation time, located near the coating/substrate interface. In addition, the coating was divided into the following two parts: a thick outer layer and a thin interface layer, and there are some microstructure defects in the coating (crack and pore). However, for pre-anodized, the sample of A1 to have appeared small cavity defects (Figure 8c, A1) and the samples of A2 and A3 were broken down and appeared to have fragmentation thinning (Figure 8b,c, A2 and A3) during oxidation 1 and 2 min. It is due to the GL breakdown and high-temperature sparking [36]. Additionally, then, the films of pre-anodized were transformed into MAO coatings after 5 min of oxidation and the coating’s growth of pre-anodized such as A0 during 5–20 min of oxidation. It indicates that pre-anodized film can accelerate the reaction in the early stage. In addition, all pre-anodized revealed the greatest changes of thickness between 5 and 10 min, tending to stabilize after 10 min. It indicates that the pre-anodization can accelerate the reaction to a certain extent, with a limit. The limit may be associated with the type of pre-anodized film or the ability to store electrons. Meanwhile, the thickness of MAO coatings of pre-anodized was obviously thicker than A0 after 10 min of oxidation (Figure 8). It indicates that pre-anodized can increase the thickness of coatings in agreement with the results of the evolution process of MAO spark (Figure 1b). The cavity defect in the coatings was increased with oxidation time.

### 3.6. Corrosion Evaluation

As shown in Figure 9, the coatings without pre-anodized A0 show more severe corrosion after immersing in a 3.5 wt% NaCl solution for 168 h compared to pre-anodized A1, A2 and A3. For pre-anodized, only the coating after oxidation for 5 min was revealed to have corroded, indicating that pre-anodized can improve the corrosion performance of MAO coating.

Figure 10 shows the mass loss of all samples after immersing in 3.5 wt% NaCl solution for 168 h. The mass loss of pre-anodized was markedly decreased compared to without pre-anodized. It is noteworthy that the pre-anodized coatings after 5 min oxidation showed a significant weight loss after immersing. However, the coating of pre-anodized with thicker MAO thickness at other oxidation times did not show a significant weight loss, indicating that the corrosion associated with the thickness of the MAO coatings. From Figure 2, the thickness of MAO coatings by pre-anodized is significantly thickened after 10 min of oxidation. The thickness increase in MAO coating makes the corrosive solution difficult to penetrate the substrate [27] or the blocking effect of the thicker MAO coatings postpones the deterioration of samples [41].

The coatings of pre-anodized revealed a similar microstructure (Figure 4, Figure 5, Figure 6, Figure 7 and Figure 8) and corrosion performance (Figure 9) after 10 min of oxidation. Each group picks up a sample after oxidation for 10 min for the potentiodynamic polarization scanning measurements to further evaluate the corrosion resistance. The potentiodynamic polarization curves of different samples are shown in Figure 11. Additionally, the Tafel fitting is carried out on the polarization curve. The fitting parameter results are listed in Table 4, such as R_p_ represents the polarization resistance, E_corr_ and I_corr_ are the self-etching potential and self-etching current density, respectively, and R_corr_ is the corrosion rate. Combined with the fitting results, it can be seen that the self-corrosion potential values of the coatings of A1, A2 and A3 are positively moved and R_corr_ is also decreased compared to A0, indicating that the pre-anodization could improve the corrosion resistance more than without pre-anodization. For MAO coating of pre-anodized, the self-corrosion potential of A2 reached up to −1.305 V, and its own corrosion current density was the lowest, 5.87 × 10^−6^ A⋅cm^−2^, indicating that the coating of A2 has the best corrosion ability. The corrosion resistance of the MAO coating was determined by a combination of coating defects and thickness. This may be due to the coating defects tending to decrease with pre-anodized film thickness, making the corrosive solution difficult to penetrate into the substrate, so the R_corr_ decreased [27].

### 3.7. Growth Model for MAO Coatings

Based on the study and discussions above, the MAO coating proposed growth mechanism of pre-anodized is depicted in Figure 12.

For without pre-anodized, as shown in Figure 12a, the growth of MAO coating is strongly related to electron accumulation and plasma discharge. In early-stage (Figure 12a), before discharged breakdown, an anodic oxidation (AO) first takes place at low voltages, according to the following anode reactions (in the case of aluminum metal in an alkaline electrolyte): Al→Al^3+^ + 3e^−^, 2Al^3+^ + 6OH^−^→2Al(OH)_3_→Al_2_O_3_ + 3H_2_O and 4OH^−^→2H_2_O (l) + O_2_ (g) + 4e^−^ [27]. A thin and relatively uniform anodic film grows on the surface with the increase in voltage. This growth is realized by the egress of Al^3+^ ions toward the film/electrolyte interface and the ingress of O^2-^ ions toward the metal/film interface [30]. With the increase in voltage (Figure 12b), the micro-arc spark begins to occur in the weak area of the anodic film, accompanied by stress when the voltage rises to the critical breakdown voltage and the electronics are accumulated. The anodic film is transformed into an MAO coating (Figure 12c). At a later stage (Figure 12d), the MAO coating gradually thickens with the voltage rise.

For the investigated system of pre-anodized, it can be summarized. The pre-anodization was divided into the following two cases: the thinner pre-anodized film (below 8 μm) and the thicker pre-anodized film (beyond 8 μm).

For thinner pre-anodized, in the early stage, the film pre-anodized with a higher impedance replaces the film formed in the conventional MAO process. So the samples that are thinner pre-anodized can store more electrons than the sample without pre-anodized with greater stress. As shown in Figure 12e, with the increase in voltage, the film of pre-anodized begins to have some crack defects due to the stress caused by electron accumulation when the working voltage exceeds the bearing limit of the pre-anodized film and does not reach the MAO breakdown voltage. The electrons will be released from crack defects through electronic transition, which is called the GL phenomenon. Some spark discharge begins to occur in the weak area of the pre-anodized film when the voltage rises to the critical breakdown voltage (Figure 12f). The pre-anodized film was gradually transformed into MAO film completely with the progress of the reaction (Figure 12g). In addition, because the samples of thinner pre-anodized can store more electrons than the sample without pre-anodized, the sample of thinner pre-anodized has a higher voltage and shows a thicker coating than the sample of the without pre-anodized (Figure 12h) at a later stage. This is also the reason why the energy consumption of the pre-anodization is less than that of the without pre-anodized at the same time. Besides, the thicker the coating, the better the corrosion resistance of the coating.

The thicker pre-anodized is different from the thinner pre-anodized. The thicker pre-anodized sample can store more electrons than the thinner pre-anodized sample. So the samples of thicker pre-anodized begin to reveal more crack defects and appear more obvious GL phenomenon than the thinner pre-anodized film in early-stage (Figure 12i) when the working voltage exceeds the bearing limit of the pre-anodized film and does not reach the MAO breakdown voltage. With the increase in voltage, the thicker pre-anodized with a higher impedance that stores more electrons, resulting in a higher working voltage. The higher working voltage will lead to higher temperature spark discharge, resulting in fast evaporation of the pre-anodized film [2]. So the pre-anodization film reveals broken down and fragmentation thinning when the voltage rises to the critical breakdown voltage (Figure 12j). With the progress of the reaction, the pre-anodized film gradually transforms into MAO film completely (Figure 12k). At a later stage, the sample of thicker pre-anodized will reveal a thicker coating than the sample without pre-anodized and thinner pre-anodized.

## 4. Conclusions

Under the present experimental conditions; major conclusions can be drawn as follows:The pre-anodization can change the spark of the early reaction and shorten the early reaction process of the MAO reaction, reducing energy consumption. The thickness of pre-anodized at 16 μm has the best energy-saving effect (2.78 kW·h·m^−2^μm^−1^) after oxidation for 10 min, saving 47%;All the pre-anodization appeared to be a breakdown by GL when the working voltage approached the breakdown voltage. The thicker pre-anodized film (beyond 8 μm) will be locally lost from the aluminum surface and appear to fragment thinning, due to the high-temperature sparking;The γ-Al_2_O_3_ phase content after pre-anodized is significantly increased in MAO coatings, associated with an increase in the reaction energy by pre-anodized that enhanced the local transient pressure as well as the temperature to promote the formation of γ-Al_2_O_3_ phases;After pre-anodized, the corrosion resistance of MAO coating was significantly improved. Better corrosion resistance was obtained of the pre-anodized compared to the without pre-anodized after 10 min oxidation, the corrosion rate was reduced by an order of magnitude;On the basis of the pre-anodized film, it is believed that the corrosion resistance of the MAO coating can be further improved by changing the electrical parameters or the composition of the electrolyte.

## Figures and Tables

**Figure 1 materials-15-05221-f001:**
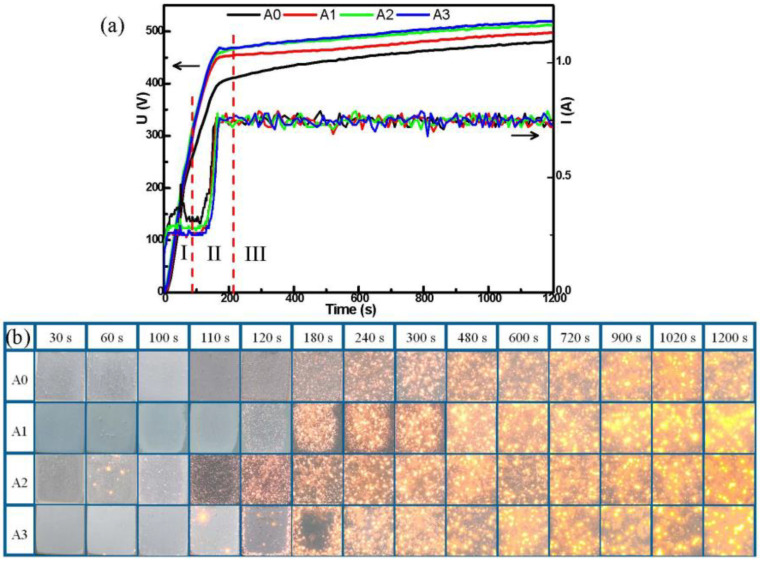
(**a**) U-T and I-T curves, (**b**) sparks pictures in the MAO process.

**Figure 2 materials-15-05221-f002:**
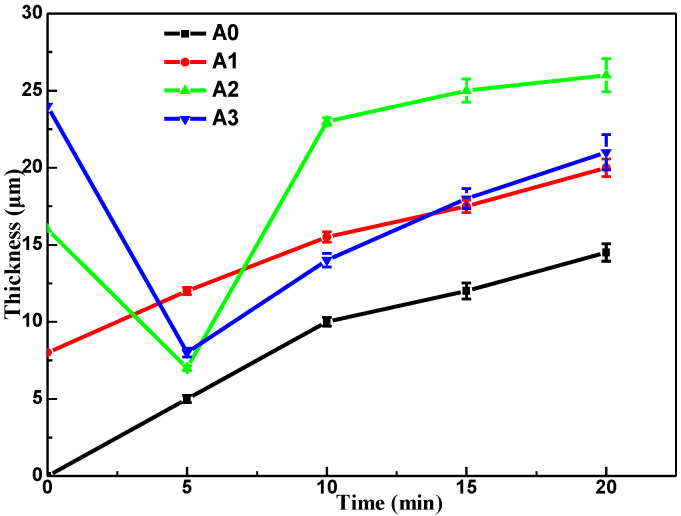
Changes in the coating thickness with the MAO treatment time.

**Figure 3 materials-15-05221-f003:**
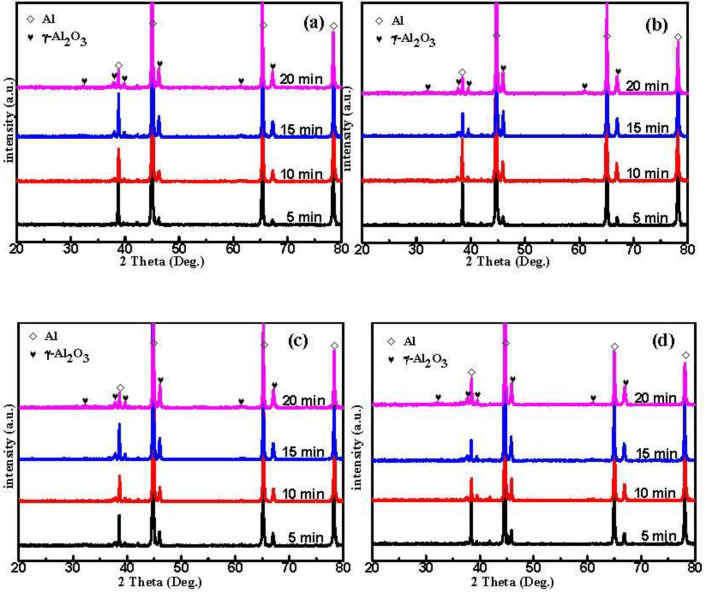
XRD patterns of (**a**) A0, (**b**) A1, (**c**) A2, (**d**) A3.

**Figure 4 materials-15-05221-f004:**
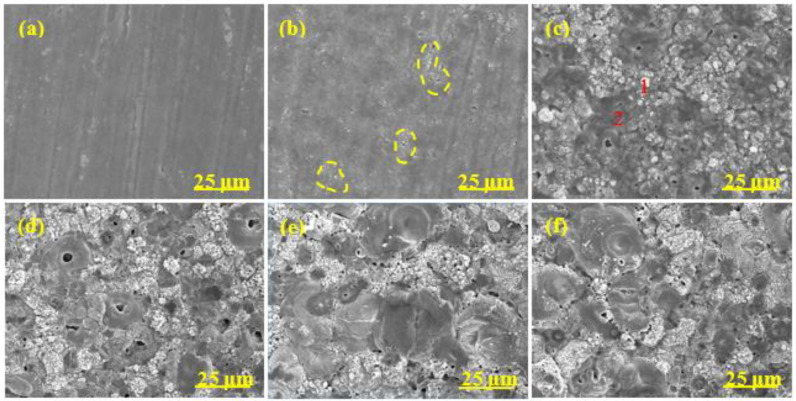
Surface morphology of MAO coatings of A0 at different time. (**a**) 1 min; (**b**) 2 min; (**c**) 5 min; (**d**) 10 min; (**e**) 15 min; (**f**) 20 min.

**Figure 5 materials-15-05221-f005:**
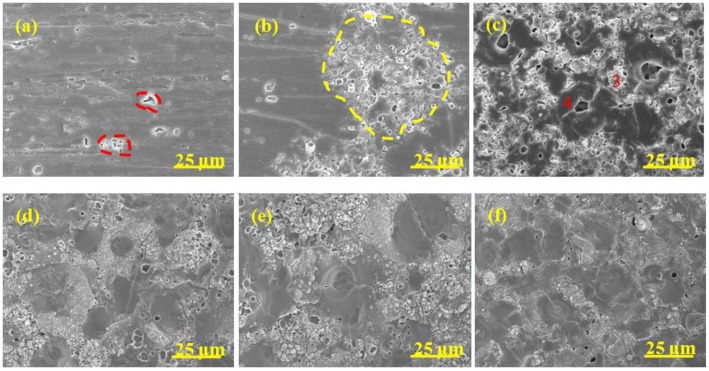
Surface morphology of MAO coatings A1 at different time. (**a**) 1 min; (**b**) 2 min; (**c**) 5 min; (**d**) 10 min; (**e**) 15 min; (**f**) 20 min.

**Figure 6 materials-15-05221-f006:**
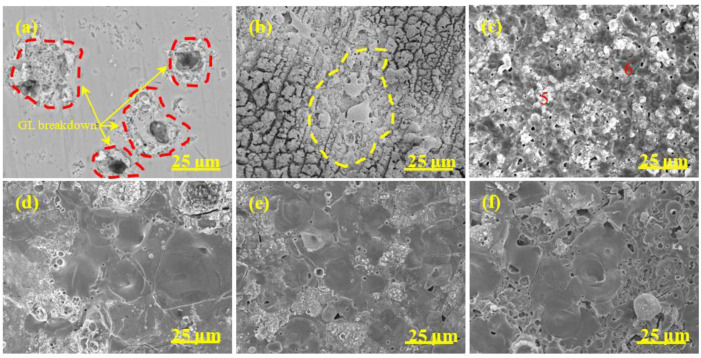
Surface morphology of MAO coatings A2 at different time. (**a**) 1 min; (**b**) 2 min; (**c**) 5 min; (**d**) 10 min; (**e**) 15 min; (**f**) 20 min.

**Figure 7 materials-15-05221-f007:**
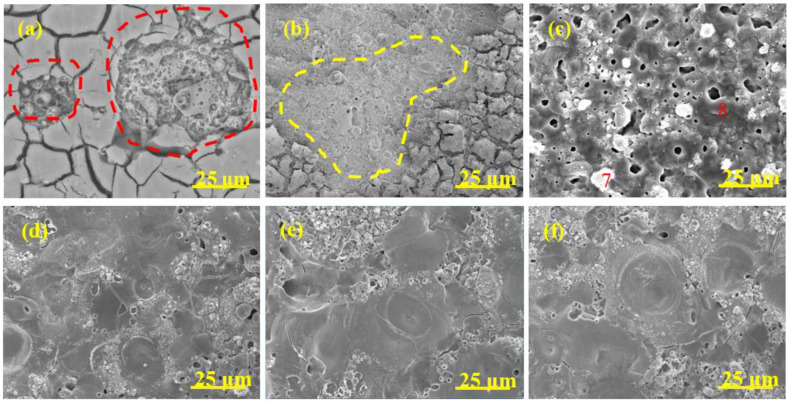
Surface morphology of MAO coatings A3 at different time. (**a**) 1 min; (**b**) 2 min; (**c**) 5 min; (**d**) 10 min; (**e**) 15 min; (**f**) 20 min.

**Figure 8 materials-15-05221-f008:**
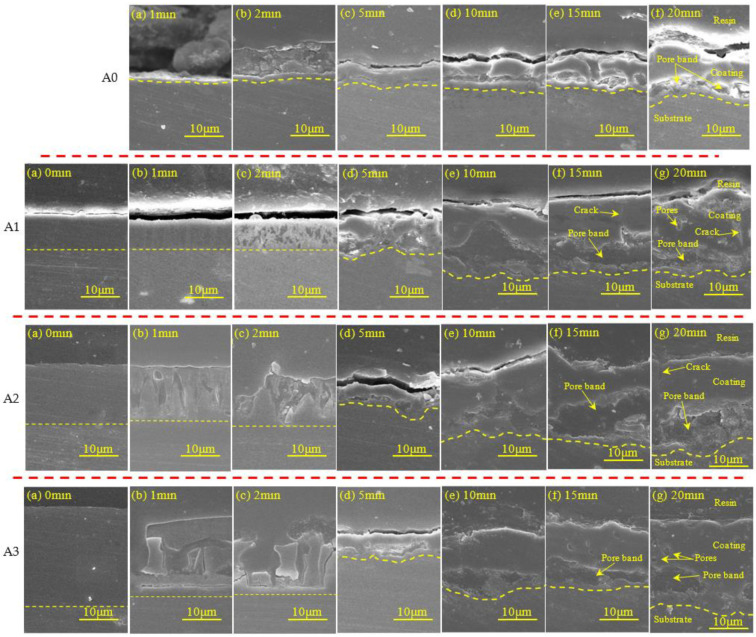
Cross-sectional changes of MAO coatings for different samples A0, A1, A2 and A3 at different time.

**Figure 9 materials-15-05221-f009:**
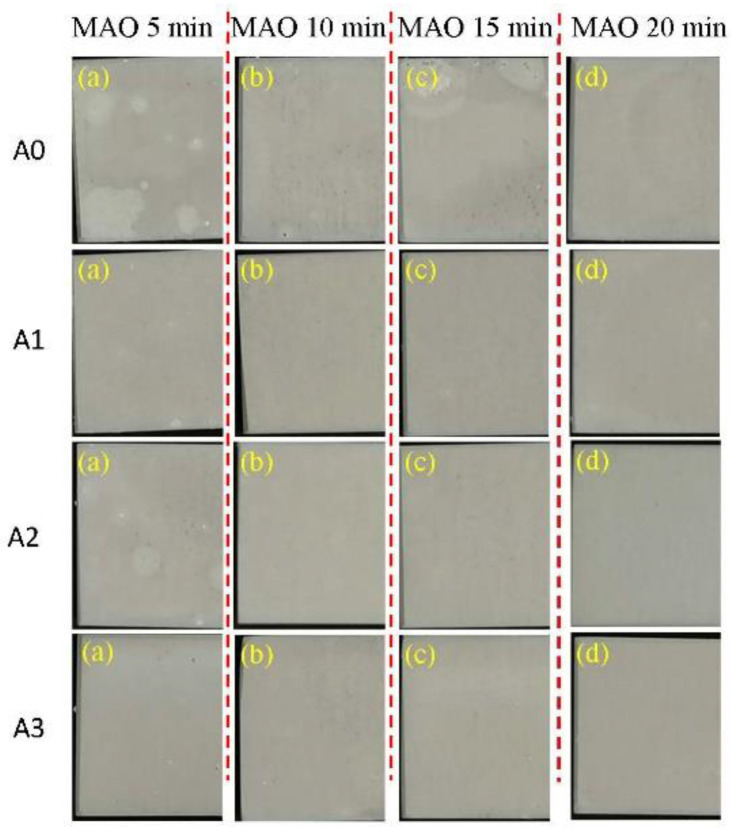
The images of A0, A1, A2 and A3 after immersion in 3.5% NaCl for 168 h.

**Figure 10 materials-15-05221-f010:**
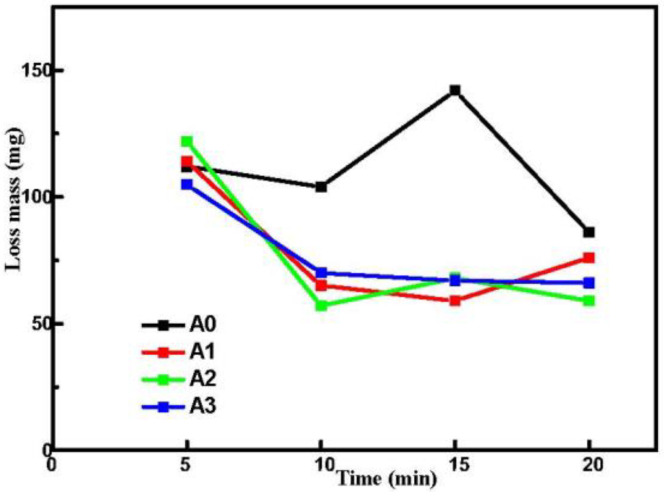
The mass loss of A0, A1, A2 and A3 after immersion in 3.5% NaCl for 168 h.

**Figure 11 materials-15-05221-f011:**
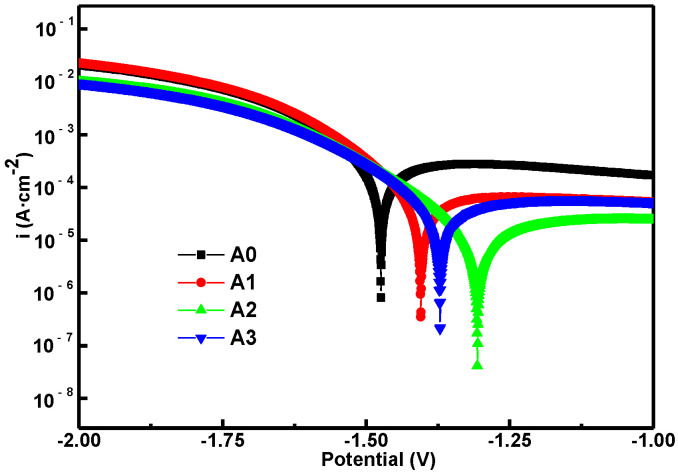
Potentiodynamic polarization curves of A0, A1, A2 and A3.

**Figure 12 materials-15-05221-f012:**
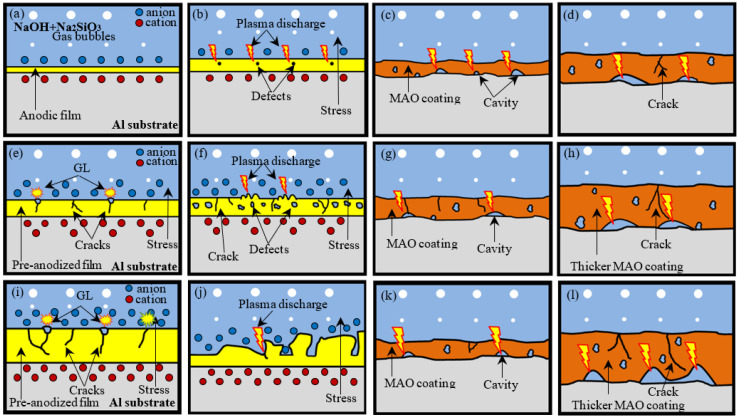
The growth mechanism model of the MAO coating process after pre-anodized: (**a**–**d**) the without pre-anodized; (**e**–**h**) the thinner pre-anodized film; (**i**–**l**) the thicker pre-anodized film.

**Table 1 materials-15-05221-t001:** Energy consumption of ρ (kW·h·m^−2^·μm^−1^).

Sample	5 min	10 min	15 min	20 min
A0	3.48	5.29	8.24	8.94
A1	2.02	4.04	5.80	7.09
A2	3.53	2.78	4.17	5.61
A3	3.20	4.77	6.05	7.24

**Table 2 materials-15-05221-t002:** EDS analysis data of Al, O and Si at different points in Figure 4, Figure 5, Figure 6 and Figure 7.

Element at %				
Sample	Point	Al	O	Si
A0	1	36.32	44.85	18.83
	2	64.56	30.26	5.18
A1	3	61.70	26.20	12.10
	4	55.65	38.08	6.27
A2	5	18.23	49.24	32.53
	6	61.46	32.74	5.80
A3	7	55.68	32.95	11.37
	8	46.11	44.14	9.75

**Table 3 materials-15-05221-t003:** EDS spectra corresponding elements scan analyses of MAO coatings in Figure 4, Figure 5, Figure 6 and Figure 7.

Element at %				
Coating	Time (min)	Al	O	Si
A0	5	53.94	38.07	7.99
	10	49.64	39.85	10.51
	15	44.37	42.40	13.23
	20	41.90	40.07	18.03
A1	5	49.31	39.40	11.29
	10	45.75	39.56	14.69
	15	41.95	39.62	18.43
	20	38.89	39.47	22.64
A2	5	52.46	38.33	9.21
	10	43.88	38.16	17.96
	15	41.53	39.01	19.46
	20	40.61	38.13	21.26
A3	5	50.69	39.91	9.40
	10	48.01	39.53	12.46
	15	44.78	39.79	15.43
	20	43.03	38.94	19.03

**Table 4 materials-15-05221-t004:** Parameters derived from polarization curves of A0, A1, A2 and A3.

Sample	E_corr_ (V)	I_corr_ (A·cm^−2^)	R_corr_ (mm·a^−1^)	R_p_ (Ω·cm^2^)
A0	–1.473	9.73 × 10^−5^	0.9535	185.08
A1	–1.403	2.61 × 10^−5^	0.2558	689.79
A2	−1.305	5.87 × 10^−6^	0.0576	3065.8
A3	–1.371	1.62 × 10^−5^	0.1587	1112.3

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
