# Peer review of "Effect of Pre-Anodized Film on Micro-Arc Oxidation Process of 6063 Aluminum Alloy"

_materials, 2022, doi:10.3390/ma15155221_

Round 1
Reviewer 1 Report
Pls see the attached file.

Author Response
Responses to Reviewer’ comments and the description of revisions in the revised manuscript
We would sincerely thank the Reviewers for their time and effort in carefully reading our manuscript (ID: materials-1800651) and in preparing the review reports. We truly appreciate their positive comments on our work, as well as for raising interesting points, which lead to the improvement of the manuscript. We have revised our manuscript accordingly, and, as a result, we believe its quality is greatly improved. The point-by-point responses to the comments are enclosed. We hope we have satisfactorily addressed all Reviewers’ concerns and questions.

Reviewer 2 Report
The paper studies the pre-anodized layer’s role in forming the micro-arc oxidation (MAO) layer to improve the corrosion resistance of aluminum alloys. The Authors decided to study the growing MAO layer's structural, morphological, and compositional properties by slowing the working voltage rate increase and analyzing the role of different pre-anodized layers (from 0 to 24um) at several process times. The Authors present many images and experimental data, with a detailed discussion leading to reasonable conclusions. However, the Authors must address several critical issues to improve the self-consistency of their findings and the overall clarity. More in detail:
1. Section 3.1: the Authors never mention the I/T evolution (Fig.1a), but this point should be addressed. The selected time points are 0, 5, 10, 15, 20 min and not 1, 2, 5, 10, 20 min (see Fig.2); however, in section 3.3 they are 1, 2, 5, 10, 15, 20 min. In Fig.1b, the pre-oxidized surfaces show some mask effects, with sparks starting first on the border (A3) or in the middle (A1), while sparking on A2 is more homogeneous. This is probably due to some issues with the pre-oxidizing procedure. Surprisingly, A2 shows the best results in terms of thickness and energy consumption (Section 3.2); is this a coincidence? The Authors should comment on this issue.
2. Section 3.3: the Authors claim, “Also, the main phase of all sample coatings were gamma-Al2O3 and alfa-Al2O3,…” but at the end of pag.9, they claim “It is worthy to note that the all MAO coatings in the present study show the absence of diffraction peaks of alfa-Al2O3”. So, is alfa-Al2O3 present or not in the XRD diffraction pattern? In Fig. 3, the Authors must also label the peaks from the Al substrate.
3. Section 3.4: Figures 3, 4, 5, and 6 should be 4, 5, 6, and 7. Define at the section beginning the selected time points. EDS analysis shows the presence of Silicon in the A0 sample, quite surprising as 6063 aluminum alloy has <0.5% of this element. The Authors should comment on this issue. The differences in stoichiometry shown in Table 3 should be better described.
4. Section 3.5: The Authors did not comment on Table 3 (page 17) results.
5. Conclusions: this is a summary of the proposed work and results; however, the Authors should also briefly describe the proposed approach and suggest further (possible) implementations to improve the achievement of the best corrosion layer.
6. Define acronyms and symbols, e.g., GL (in the abstract), Rpn (pag.2),
7. Fig.1a: left and right labels must be “U (V)” and “I (A)”, bottom label “Time (s)”
8. Fig. 3, 4, 5, 6 (numbers must be changed): the caption must describe the different process times and the meaning of the red numbers.
9. Table 3, page 17, should be Table 4 and correspondingly all other following tables.
10. Fig.7: improve figures description in the caption (the interface line, the numbers…)
11. Some phrases are not self-consistent or challenging to understand. E.g., pag.2, “The way…treatments”; pag.6, “It may be a duo to the film…MAO process”, “Make the working…same time”; pag.7, “Select a certain time point…” should be “We selected a specific time point…”; pag.8, “The pre-anodized can significant reduction…”; pag 18, “…were revealed a more corroded…”
Author Response

(The authors gave the same response as above.)

Round 2
Reviewer 1 Report
Please see the attached file

Author Response
We would sincerely thank the Reviewers for their time and effort in carefully reading our manuscript (ID: materials-1800651) and in preparing the review reports. We truly appreciate their positive comments on our work, as well as for raising interesting points, which lead to the improvement of the manuscript.
We again revised the manuscript accordingly, and, as a result, we believe its quality is greatly improved. The point-by-point responses to the comments are enclosed. We hope we have satisfactorily addressed all Reviewers’ concerns and questions.

Reviewer 2 Report
The Authors accepted most of the proposed suggestions, addressing and clarifying the most important issues. Paper can be published in this form.
Author Response
We sincerely thank the reviewers for taking the time and effort to peruse our manuscript (ID:materials-1800651) and prepare the review report. We greatly appreciate their positive comments on our work, as well as interesting points that have led to improvements in the manuscript. Thanks again to the reviewers for acknowledging our work. We hope that we have satisfactorily addressed all reviewer concerns and questions.